# A Manufacturing Process Simulation of Toughened Cyanate-Ester-Based Composite Structures with Respect to Stress Relaxation

**DOI:** 10.3390/ma15196675

**Published:** 2022-09-26

**Authors:** Nicolas Gort, Fabian Schadt, Martin Liebisch, Christian Brauner, Tobias Wille

**Affiliations:** 1Institute of Polymer Engineering, FHNW University of Applied Sciences and Arts Northwestern Switzerland, Klosterzelgstrasse 2, 5210 Windisch, Switzerland; 2Institute of Composite Structures and Adaptive Systems, German Aerospace Center, Lilienthalplatz 7, 38108 Braunschweig, Germany

**Keywords:** composite materials, cyanate ester, toughening, high-temperature application, process-induced deformation/stress, viscoelastic, relaxation

## Abstract

The objectives of this study were to experimentally determine the effects of the stress relaxation of a cyanate-ester-based composite, derive and integrate constitutive equations into commercial FEM software, and apply this approach to understand the formation of residual stress in a typical aerospace structure—namely, a stiffened panel. In preliminary studies, a cyanate-ester-based composite with increased fracture toughness for high-temperature applications was developed. High curing temperatures up to 260 °C will inevitably lead to high process-induced stresses. To assess the magnitude of impact on the development of internal stresses, the relaxation behavior of the neat resin was measured and characterized. The system was toughened, and the effect of stress relaxation increased as the temperature got closer to the glass transition temperature of the toughener, which was approximately 240 °C. With the use of an incremental linear viscoelastic model, the relaxation behavior was integrated into a process model with a holistic approach. A stiffened panel was manufactured and used as the validation use case. The displacement field was validated with an optical 3D measuring system, and good agreement was found between the simulated and experimental results. The maximum difference between the elastic and the viscoelastic solution was found to be 15%. Furthermore, the stress magnitude in the transverse material direction resulted in a more critical value higher than the material strength.

## 1. Introduction

A distinctive feature of composite materials is that the resulting material properties can be designed by selecting different reinforcing and matrix materials. For applications with increased requirements on temperature resistance thermoset materials, such as Bis-Maleimide (BMI), Cyanate Esters (CE), or Polyimide (PI), are suitable. These materials are used in the aerospace industry, including in turbine housing, exhaust systems, and auxiliary power units. The ability to resist high temperature loads can be estimated by the value of the glass transition temperature (*T_g_*) of the material of interest. A comparison of commercially available high-temperature-resistant materials for structural applications above 250 °C operational temperatures is provided [1] in Figure 1.

Notably, CE-based composites showed comparably high *T_g_* values and hence are of potential interest for applications with high thermal loads. These composites are currently used in areas, such as high-temperature adhesives and advanced composite matrices in the aviation, aerospace, and automotive industries [2,3,4]. A unique combination of properties, including high temperature stability, high chemical resistance, low moisture uptake, and a low dielectric constant in the cured state, as well as low viscosity in the uncured state, has led to the use of these materials in low-volume high-performance applications. However, the widespread use of such composites is unfortunately limited in many applications by their inherent brittle behavior due to their high cross-linked densities.

This challenge was faced in the European research project SuCoHs (Sustainable and Cost Efficient High-Performance Composite Structures demanding Temperature and Fire Resistance, Funding No 769178). The aim of this project was to enable the commercial use of polymer composite materials in regions with high thermal loads. To make the material tougher and thus suitable for use as a matrix material in primary structures, a method was developed using thermoplastic polyether sulfone (PES) [5,6]. Amongst other possibilities [7], the reason for choosing PES as a toughener was its ability to create crosslinks with the CE as explained in [8].The result is a novolac-based CE material with a *T_g_* at 390 °C. Despite this high *T_g_*_,_ the developed system allowed autoclave curing at 180 °C; nevertheless, it requires a post-curing temperature of 260 °C [5]

Based on these extraordinary high temperatures required for curing and post-curing, process-induced deformations and residual stress might become critical. One remaining insecurity is the temperature-dependent behavior, especially the viscoelastic behavior occurring at high temperatures. Therefore, the objectives of this study were to experimentally determine the effects of stress relaxation, derive and integrate constitutive equations into a commercial FEM, and apply this approach to understand the formation of residual stress in a typical aerospace structure, namely a stiffened panel. As a demonstration of the capability of the developed material, the material was integrated into the production of a demonstrator part for mechanical testing. The manufactured panel can be seen in Figure 2. Further details on the panel’s manufacturing, as well as the process simulation, are outlined in Section 4.

The starting point of this study was a comparison of the elastic modulus of pure resin and a toughened version of the material, characterized by dynamic mechanical analysis (DMA). In the second step, the time-dependent viscoelastic behavior was characterized by relaxation tests at different temperatures.

In the second part of this study, the resulting parameters were taken as input values for the existing holistic approach to analyze process-induced deformations and residual stresses [9,10]. This approach is suitable for transient sequential coupled thermo-mechanical analysis. In this study, the stresses were compared between the viscoelastic solution and the elastic solution. In addition, the process-induced deformation was compared to the final deformation of the manufactured demonstrator.

While the viscoelastic phase separation of thermoset/thermoplastic blends (also for the case of cyanate-ester resin [11]) has been investigated, no measurements have yet been found for the temperature-dependent macroscopic relaxation behavior. The use of polymers in high-temperature applications is the subject of current research, which is why there is no comparable literature on this specific topic. The novelty of this study is not only the measurement of temperature dependent macroscopic relaxation behavior of toughened cyanate-ester resin but also the assessment of its effects on the residual stresses and deformation of a larger component.

## 2. Materials and Methods

This section characterizes the thermomechanical viscoelastic behavior of resin via dynamic mechanical analysis (DMA). First, the resin samples were used to determine the storage modulus over temperature. In the second step, relaxation tests were performed as the starting point to derive a viscoelastic material model. Characterization of the single key aspects yielded constitutive equations and parameters to model the material behavior.

### 2.1. Materials

The present study concentrates on a novolac cyanate-ester-based resin (CE, PT30) supplied by Arxada (Basel, Switzerland). This resin is intended for manufacturing aircraft composite structures. As a toughener, polyethersulfone (PES) from Sumitomo (Tokyo, Japan) was used.

The CE material was modified to increase the fracture toughness [6]. A comprehensive study was performed to find a suitable thermoplastic additive that increases fracture toughness without decreasing the final *T_g_*. The incorporation of thermoplastics into such networks has emerged as a promising approach to improve toughness, especially when high values of the elastic moduli and *T_g_* are required. In this context, we investigated the morphology and thermal and mechanical properties of the CE-PES blends, depending on the molecular weight and content of the thermoplastic toughener [8]. Increases in the molecular weight of the thermoplastic toughener led to phase separation, which caused a reduction in the *T_g_* of the blend towards that of PES (approximately 240 °C). Depending on the content of PES, the minor drop in the storage modulus could allow such systems to be used in high-temperature composite applications. This study found the highest toughening effect at 20 parts per hundred rubber (phr) (132% increase) but concluded that the best compromise of the *T_g_* and toughening effect could be observed at a concentration of 15 phr. This was chosen to be the baseline material for the relaxation tests and modeling in this article.

Based on the developed matrix material, a unidirectional thin ply prepreg (t = 60 µm) was developed for use in advanced fiber placement (AFP) on a Coreolis AFP machine [12]. The prepreg was manufactured by the company North Thin Ply Technology (Yverdon, Switzerland) (NTPT) using T800 fibers provided by Toray (Tokyo, Japan). The prepreg was manufactured with a 67 g/m^2^ aerial weight and a nominal fiber volume content of 55%. To cure the material, the material was heated at a heating rate of 1 °C/min and held constant for 2 h at 120, 180, and 260 °C each.

### 2.2. Measurement of the Temperature-Dependent Elastic Behavior

The *T_g_* and storage modulus of the neat resin system were measured using a TA Instruments (New Castles, WY, USA) Q800 dynamic mechanical analyzer (DMA). The measurements were performed using a three-point bending setup. A heating rate of 5 °C/min from room temperature to 400 °C was applied. The specimens were cut with a diamond blade saw (Compcut, Holsworthy, United Kingdom) and polished to dimensions of 25 × 10 × 2 mm. The *T_g_* was defined as the temperature corresponding to the onset of the storage modulus curve. In order to demonstrate the effects of different concentrations of PES to the onset of the drop in the storage modulus curve, concentrations of 5 to 20 phr were measured in addition to the pure CE.

### 2.3. Measurement of Temperature-Dependent Relaxational Behavior

To measure the stress-relaxation abilities of the neat resin system, the same setup was used for a sweep of relaxation tests at different temperatures. This method was chosen due to its possibility to characterize relaxation dynamically in the time domain because of the slow characteristics of the load change on the material during the manufacturing process.

The temperature range was set in descending order from 350 to 40 °C in steps of 25 °C to prevent post-curing effects. At each specific temperature, a sample in a three-point bending test was dislocated (1% strain) and held constant for one hour. Meanwhile, the resulting stress was recorded. A preload of 0.1% strain was used, and the samples for this measurement were manufactured to dimensions of 25 × 10 × 2 mm. Calculation of the relaxation modulus was performed as follows:(1)$$E\left(t\right)=K_{S}\left(t\right)\frac{L^{3}}{6I}[1+\frac{6}{10}\left(1+v\right){\left(\frac{h}{L}\right)}^{2}]$$
where *E*(*t*) denotes the relaxation modulus, *L* is the sample length, *h* is the sample thickness, *I* is the moment of inertia, *v* is Poisson’s ratio, and *K_S_*(t) is the measured relation between force and deflection.

### 2.4. Characterization of the Temperature-Dependent Linear Viscoelastic Behavior

A common method to describe the relaxation behavior of a linear viscoelastic material is the use of a rheological model called the generalized Maxwell model, also known as the Wiechert model [13]. This model consists of a number of so-called branches that are connected in parallel, each consisting of a spring and dashpot (Figure 3).

To characterize the performed relaxation measurements, the response of the generalized Maxwell model consisting of *M* = 2 branches was fit to the measurement data by varying the parameters *E* and *η* using a linear regression algorithm. The result was a value for each element of the rheological model and each measured temperature.

To derive a continuous description over temperature, a custom parametrized expression was subsequently fit.

## 3. Results

In this section, we first present the results of the measurements of elastic behavior followed by those of the relaxation measurements. Lastly, the relaxation measurement data are used to calibrate the customized approach.

Investigating the elastic behavior showed the effects of different contents of thermoplastic additives. The studied thermoplastic material, PES, had a *T_g_* between 220 and 230 [14]. Without additives, the CE had a final *T_g_* of up to 405 °C. By increasing the amount of additives, a new *T_g_* emerged, which is visible in the second drop between 225 and 250 °C. This effect is clearly visible in the plot of the storage moduli shown in Figure 4. The absolute measured values of the *T_g_* depending on the PES content are given in Table 1.

In the next step, the relaxation behavior was studied as described above. In Figure 5, the measured relaxation modulus is presented. Here, an increase in the instantaneous modulus under a lower temperature is visible, as well as a decrease in the relative amount of stress relaxation.

Next, the measured relaxation curves were fitted with the response of the generalized Maxwell model with two branches. The result of the linear regression fit was a value for each element of the rheological model for each discrete temperature. The resulting values are shown in Figure 6. Here, the model parameters are expressed in terms of the decay time $\rho_{i}$ (a): branch 1, (c): branch 2 and the spring constant $E_{i}$ compared to the sum of all spring constants $E_{0}$ (b): branch 1, (d): branch 2. It can be seen that the value of the springs connected to the branch with the dashpot show a step at approximately 200 °C.

The value for each element of the model is aimed to be described in a continuous manner. While the relaxation times are modeled with a polynomial approach, the springs of the branches are modeled with a superposition of an exponential curve and a step function in order to represent the step at 200 °C:(2)$$\rho_{1}=r1$$
(3)$$\rho_{2}=r2\;\cdot \;T{\left(K\right)}^{4}+r3\;\cdot \;T{\left(K\right)}^{3}+r4\;\cdot \;T{\left(K\right)}^{2}+r5\;\cdot \;T\left(K\right)+r6$$
(4)$$\frac{E_{1}}{E_{0}}=\min\left[50,\;r7\;\exp\left(r8\;\cdot \;T\left(K\right)\right)+r9-\frac{r10}{1+\exp\left(\frac{T\left(K\right)-r11}{r12}\right)}\right]$$
(5)$$\frac{E_{2}}{E_{0}}=\min\left[50,\;r13\;\cdot \;\exp\left(r14\;\cdot \;T\left(K\right)\right)+r15-\frac{r16}{1+\exp\left(\frac{T\left(K\right)-r17}{r18}\right)}\right].$$

These continuous approaches were fit to the discrete values using a linear regression algorithm. Although several solutions are possible, one of them is sufficient, which maps the behavior for the application with sufficient accuracy. The resulting parameter values for Equations (2)–(5) are given in Appendix A (Table A1). For verification, the relaxation modulus described with the rheological model and the continuous element description are compared with the initial measurement data. Figure 7 indicates good agreement between the modeled relaxation modulus and the measured relaxation behavior.

## 4. Implementation and Application on a Structural Level

The objective of this study was also to apply the developed approach on a structural level to understand the formation of process-induced deformations and residual stresses for a typical aerospace structure. The developed thin-ply material was used to produce a stiffened panel (see Figure 2). This material was then used for the comparison with the process simulation in the following section. The following layup was chosen in the panel (see Table 2).

The stacking was performed via hand layup, and the curing was performed in a standard autoclave (IPROP, Parma, Italy) at 180 °C under 5 bar autoclave pressure (total process time, 480 min). The post curing was done free-standing at 260 °C (total process time, 480 min) in a convection oven.

### 4.1. Simulation Strategy

The structure of the manufacturing process simulation consisted of a sequentially coupled thermochemical-mechanical transient analysis.

First, the thermochemical part included calculating the temperature and degree of cure for the dependent variables, as well as the *T_g_* as a function of these two parameters. The thermal module was coupled with a differential equation that describes the curing reaction, as outlined by L. Amirova et al. [4]. It consists of a separated catalytic and autocatalytic part as proposed by [16]. To consider the decrease of the reaction speed towards the end of the curing reaction, the model was modified with an additional factor [17].

In the following mechanical part, the matrix stiffness was calculated based on the actual degree of cure and the temperature. In this way, Brauner [9] proposed a model to link the current *T_g_* of the resin with rheometer measurement data. Together with the elastic fiber properties, this model can be used to calculate the homogenized layer properties and structure of the orthotropic stiffness matrix. Next, the inelastic strain consisting of thermal shrinkage and chemical shrinkage was calculated by multiplying the change in temperature using the coefficient of expansion and the change in the degree of cure via the coefficient of chemical shrinkage. In the last step, these inelastic strains were included in the calculations of the displacement and stress field of the structure.

In the end, the stress relaxation of the material was considered based on an incremental description of the response of a generalized Maxwell model using a state variable to calculate the relaxed stress at a certain point in time, depending on the loading history, which was developed by [18] and applied for example in [19]. This process can be explained as follows:

The general stress-strain relationship for an orthotropic linear viscoelastic material can be expressed with the following constitutive equation:(6)$$\sigma_{ij}\left(x_{k},\xi \right)=\int_{0}^{\xi }C_{ijkl}\left(x_{k},\xi -\xi^{\prime }\right)\frac{\partial \varepsilon_{kl}\left(x_{k},\xi^{\prime }\right)}{\partial \xi^{\prime }}d\xi^{\prime }$$

This equation is called a convolution integral and respects the history of the material by assuming that each response to a loading event can be superposed to all other responses. The generalized Maxwell model has the advantage that its response can be expressed in terms of a Prony series. Each element of the series consists of an exponential decaying term.

By assuming that the behavior of the elements of the stiffness tensor can be represented in terms of a generalized Maxwell model with *M* branches, the elements can be expressed as
(7)$$C_{ijkl}\left(x_{k},\xi_{n+1}-\xi^{\prime }\right)=C_{ijkl_{\infty }}+\sum_{m=1}^{M_{ijkl}}C_{ijkl_{m}}e^{-\left(\xi_{n+1}-\xi^{\prime }\right)/\rho_{ijkl_{m}}}.$$

This description was inserted into Equation (6) and converted into an incremental form based on the model in [18]. The actual stress is a summation of the stress increment added to the stress of the previous time step:(8)$$\sigma_{ij\left(t+\Delta t\right)}=\sigma_{ij\left(t\right)}+\Delta \sigma .$$

This stress increment takes into account the thermal and chemical strain and stress increment $\Delta \sigma_{ij}^{R}$, which considers stress relaxation. For small time steps, the immediate increment can be understood as the elastic increment and $\Delta \sigma_{ij}^{R}$ as the viscoelastic part:(9)$$\Delta \sigma_{ij}=C_{ijkl}^{\prime }\Delta \varepsilon_{kl}-\beta_{ij}^{\prime }\Delta \Theta +\Delta \sigma_{ij}^{R}$$

Contrary to [18], the temperature load considered ΔΘ was integrated into the mechanical load $\Delta \varepsilon_{kl}$. The resulting set of equations to be implemented was then written as follows:(10)$$C_{ijkl}^{\prime }\equiv C_{ijkl_{\infty }}+\frac{1}{\Delta \xi }\sum_{m=1}^{M_{ijkl}}\eta_{ijkl_{m}}\left(1-e^{\left.-\Delta \xi /\rho_{ijkl_{m}}\right)}\right.$$
(11)$$\Delta \sigma_{ij}^{R}=-\sum_{k=1}^{3}\sum_{l=1}^{3}\sum_{m=1}^{M_{ijkl}}\left(1-e^{-\Delta \xi /\rho_{ijk_{m}}}\right)S_{ijkl_{m}}\left(\xi_{n}\right)$$
(12)$$\left.S_{ijkl_{m}}\left(x_{k},\xi_{n}\right)=e^{-\Delta \xi /\rho_{ijkl}m}S_{ijkl_{m}}\left(x_{k},\xi_{n}-\Delta \xi \right)+\eta_{ijkl_{m}}R_{kl}\left(1-e^{-\Delta \xi /\rho_{ijk_{m}}}\right)\;\right)$$
where *R* is defined as
(13)$$R_{kl}\equiv \Delta \varepsilon_{kl}/\Delta \xi .$$

The variable $S_{ijkl_{m}}$ can be understood as a state variable that needs to be stored in the previous time step and forwarded to the current step. In this way, each branch *m* of the Maxwell model for each stress tensor index *ij* needs a state variable. For the general orthotropic case, $C_{ijkl}^{\prime }$ is a 6 × 6 matrix. Since each entry needs *M* state variables, there are *M* × 36 state variables. In other words, each index of the stress relaxation increment consists of the sum of all M branches of a specific index *ijkl* and the sum over *kl* in order to obtain index *ij*.

Since the impact on $S_{ijkl_{m}}$ at a specific time step due to an increase in stress is coupled to the temperature-dependent model parameters at that time, this type of characterization is only valid for comparably slow temperature changes.

In order to describe the relaxation of the composite material out of the behavior of the measured neat resin system, a homogenization method was needed. Therefore, a micromechanical model was established, where indexes of the relaxation tensor were examined with the use of isolated load cases. In this way, only the polymer material was considered to show time-dependent behavior.

In a comparison with Figure 6, it can be seen that at curing temperatures of 120 and 180 °C, the measured relaxation of the fully cured neat resin was below 10% of the elastic modulus. Since the material at this point was still in a liquid or rubbery phase, modeling the stress relaxation with the use of fully cured resin would be too conservative. Therefore, a cure dependency of the relaxation behavior was introduced to linearly change the introduced reference temperature depending on the relation of the actual glass transition temperature and the temperatures of the uncured and fully cured states:(14)$$T_{ref}=\frac{T-Tg_{0}}{Tg-Tg_{0}}*\left(Tg_{1}-Tg_{0}\right)+Tg_{0}\;(for\;T<Tg).$$

Thereby, *T_g_*_0_ denotes the *T_g_* of the uncured material (estimated: −11 °C) and *T_g_*_1_ the *T_g_* of the fully cured material (measured above: 397 °C)

The approach described here was implemented by using user subroutines in the commercial finite element program Abaqus (6.13, Maastricht, The Netherlands). For the thermochemical part in the beginning a 20-node quadratic heat transfer element was used (DC3D20), while in the subsequent mechanical analysis a quadratic element for stress analysis with also 20 nodes was chosen (C3D20R). The same number and position of nodes in the thermochemical and mechanical analysis allows the mapping of the temperature of the previous step to the next. To model the representative panel in Figure 2, a number of 53334 elements was used for each step.

### 4.2. Simulation Results and Validation

The finite element model not only helps to assess process-induced stress development during the process but also gives insight into a wide range of parameters that might not be measurable in reality. In the first step, the thermochemical model is used to help to assess the cure state and to which temperatures the part was exposed. An example is shown in Figure 8, which shows the temperature on the surface, as well as the temperature, degree of cure, and *T_g_* in the middle of the stringer filler in the middle of the panel. Here, comparison of the temperatures at these two locations indicates that in the specific case of a thin-walled structure, the temperature overshoot due to exothermic reactions is negligible, even at the thickest section. Further, even if the degree of cure reaches approximately 75% after the first curing step, this result is not the same for *T_g_*, which only reaches approximately 100 °C—far from its final value (approximately 30% of final *T_g_*). The material changes from liquid (I) to rubbery (II) to a solid (III) phase. Notably, free-standing the post curing poses high risks for the quality of the final part because by using this cycle, the *T_g_* after the first curing cycle is low, and the material starts to soften again during heating in post curing.

Applying the prescribed structural analysis methodology yielded a displacement field, as shown in Figure 9. The maximum distortion is 4.37 mm. This distortion is located at the panel tips and is mostly due to upward bending. 

The as-built configuration was captured by DLR (Braunschweig, Germany) with an optical 3D measuring system ATOS by GOM (Braunschweig, Germany). This system generates an STL file that can be compared to the as-planned (undeformed) geometry via manual overlay so that the maximum deviation between the geometries can be minimized. The global comparison was done similar to previous studies [20] using MeshLab and the built-in Hausdorff distance. The obtained result was then scaled to the measured maximum distance based on a comparison of the as-planned geometry and the STL file obtained from scanning. The resulting displacement field is shown in Figure 10.

When shaping the measurements of the demonstrator panel (see Figure 2), we used GOM ATOS to validate the numerical analysis.

It can be seen that the manual alignment of the as-planned and as-built geometry is sufficient since the maximum deviation is almost equal on both panel tips. However, the warping cannot be eliminated by better alignment to the as-planned geometry, which indicates a valid deviation from the as-planned configuration.

Since the simulated geometry is perfectly symmetric in reinforcement direction, it does not result in any warping, which in reality always happens to a certain extent. The surprisingly good agreement of the maximum simulated and measured displacement values therefore is coincidentally. Taking the mean value of the four outer corners, the maximum deflection of the produced panel would be approximately 20% lower.

In order to assess the effects of stress relaxation, the in-plane stress components at a location far from the panel edges were compared when developing the elastic and viscoelastic solution. To do so, the evaluated location at the skin laminate indicated as position 1 in Figure 9 was selected. Since the laminate was designed as quasi-isotropic, the stresses in the material coordinates were found to be the same for every layer. The development of the in-plane stress components is shown in Figure 11. *S*_11_ denotes the normal stress in the material coordinates, which refers to the fiber direction, and *S*_22_ refers to the transverse stress. A distinction was made between the viscoelastic (dashed) and elastic (solid lines) solutions. As long as the laminate was fixed in the longitudinal direction, the elastic and viscoelastic solutions were almost the same since there was hardly any stress relaxation in the fiber direction. In the transverse direction, there was a visible difference caused by the pronounced capacity for stress relaxation in this direction. Demolding led to stress redistribution since the contraction restriction caused by the fixation was taken by the fibers at this stage. This phenomenon resulted in pressure stress in the fiber direction, with hardly any visible difference in the transverse direction due to the lower stiffness exceeding a factor of 10. As a consequence of the quasi-isotropic laminate, after demolding, the stress magnitude in the longitudinal and transverse directions were of the same amount but with opposite signs.

Considering the cure dependency of the relaxation (dotted line), the viscoelastic solution showed resulting stresses that were significantly lower during the curing step, especially in the transverse direction. Additionally, the subsequent post-curing step erased part of the history such that the process-induced stresses after post-curing were caused mostly by cooling of the post-curing temperature. In this way, the difference between the elastic solution and viscoelastic solution was no greater than 10% for the least conservative assessment considering cure dependency of the relaxation.

The maximum stress magnitude observed at the end of the process was approximately 70 MPa, which was already in the order of magnitude of the material strength in the transverse direction. Notably, the linear viscoelastic solution still provided a conservative assessment of the occurring stresses since it did not include nonlinear viscoplastic effects. Especially in the liquid and rubbery state (phase I and II), the material is expected to show plastic deformation that would lead to a reduction of stress. This was not considered in the current modeling. 

Similar stress development was observed in the UD-stringer filler. The presented method was used to optimize the geometry of the stringer root section through the evaluation of different filler materials and geometric concepts [15].

## 5. Conclusions

Earlier measurements of the elastic modulus showed that an increasing amount of PES toughener led to the development of a second *T_g_* at approximately 245 °C. The increase in stress relaxation observed at a temperature of approximately 200 °C could also have been caused by the same effect—namely, the weakening of the toughening phase.

By using the incremental linear viscoelastic model, the relaxation behavior was integrated into the process model. The displacement field of the representative panel was validated with an optical 3D measuring system and found to show good agreement with the maximum deflection but slight warping in the as-built structure.

In the assessment of the development of stresses in the structure during the manufacturing process, the difference between the elastic solution and the solution when considering the relaxation behavior of the fully cured resin was found to be negligible. This is because the manufacturing process does not exceed temperatures of 180 °C until the post-curing step so that it stays below the observed increase of stress relaxation. However, the difference to the elastic solution became significant when the cure dependency of the relaxation was introduced. 

In any case, post-curing was seen to have an erasing effect on the part’s loading history such that the main part of the process-induced stresses was caused by cooling from the post-curing temperature. As a consequence, the maximum difference between elastic and viscoelastic solution at the end of the process was found to be 10%. The magnitude of macroscopic values compared to the strength of the material indicates the development of process-induced cracks in a transverse direction.

## Figures and Tables

**Figure 1 materials-15-06675-f001:**
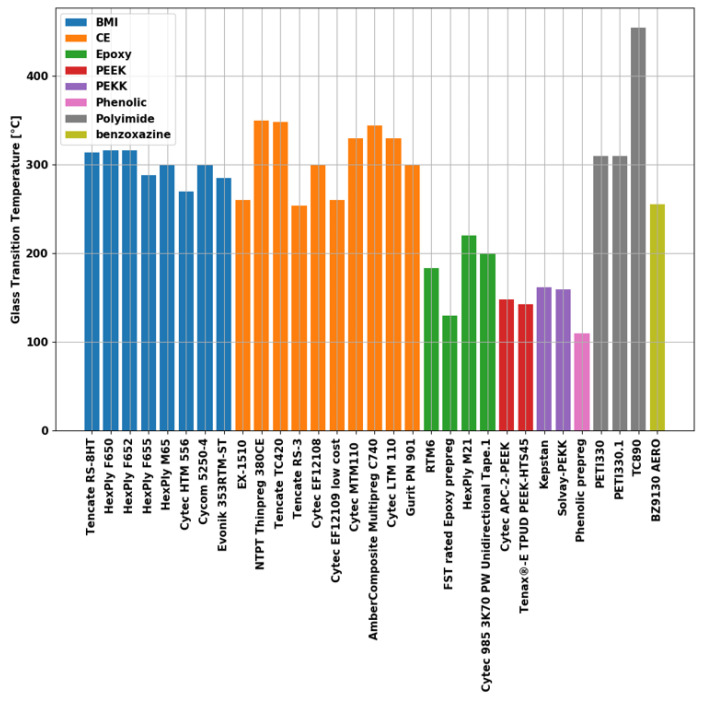
Overview to the glass transition temperature of commercially available high-temperature resin systems based on their data sheet values. Result from material screening in [1].

**Figure 2 materials-15-06675-f002:**
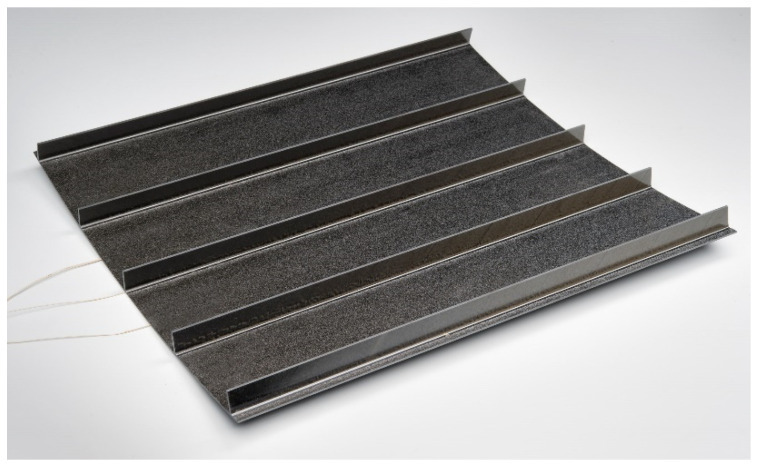
Validation structure to understand the formation of process-induced deformation and residual stresses at the structural level.

**Figure 3 materials-15-06675-f003:**
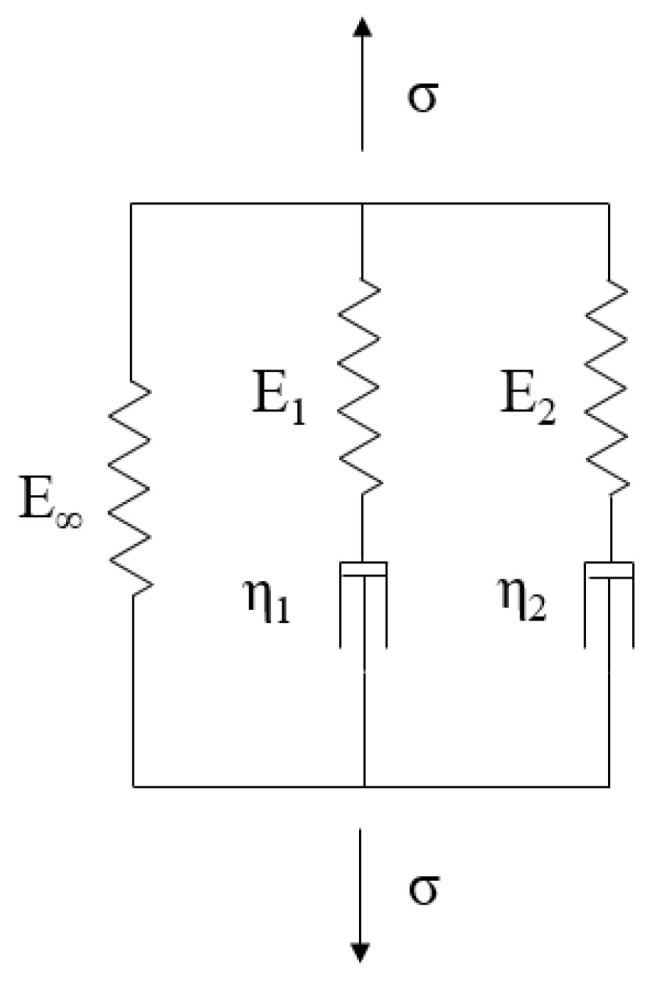
Generalized Maxwell Model or Wiechert Model [13].

**Figure 4 materials-15-06675-f004:**
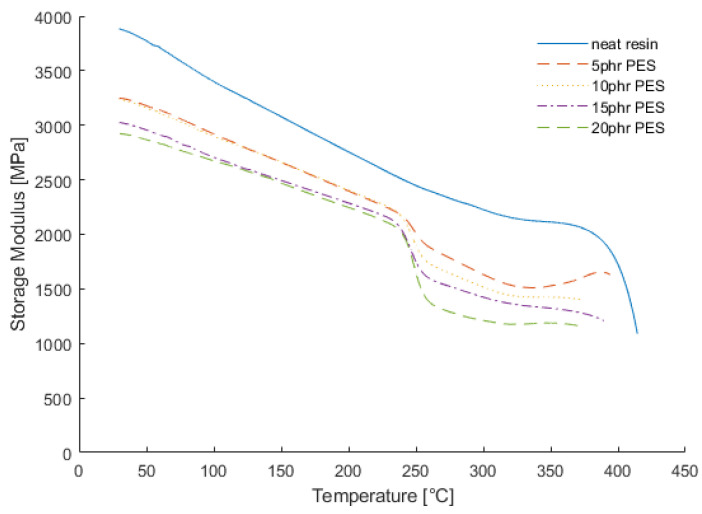
Storage modulus measured via dynamic measurements.

**Figure 5 materials-15-06675-f005:**
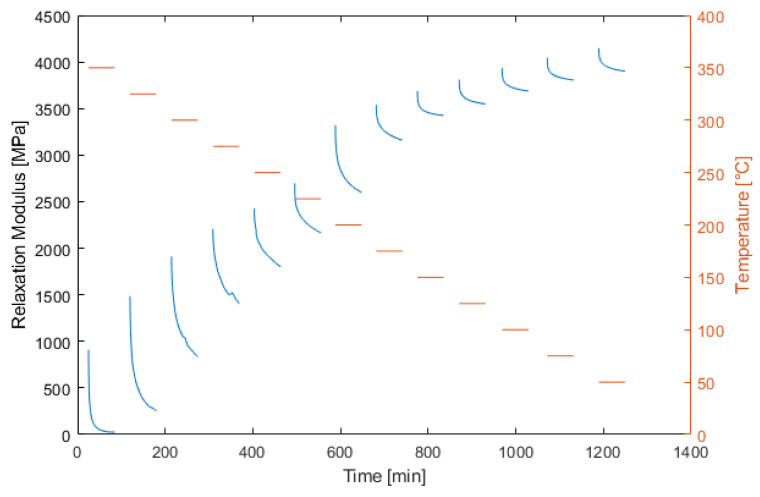
Measured relaxation modulus for a temperature sweep from 350 to 20 °C.

**Figure 6 materials-15-06675-f006:**
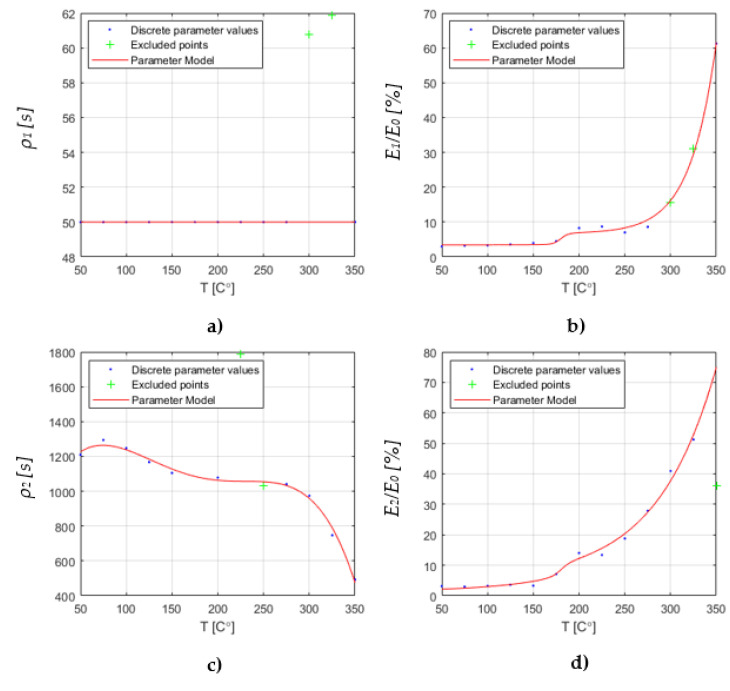
Resulting values of the element parameters of the Maxwell Model and the parametric fit for a continuous description [15]. It shows the decay time
$\rho_{i}$ (**a**): branch 1, (**c**): branch 2 and the spring constant $E_{i}$ compared to the sum of all spring constants $E_{0}$ (**b**): branch 1, (**d**): branch 2.

**Figure 7 materials-15-06675-f007:**
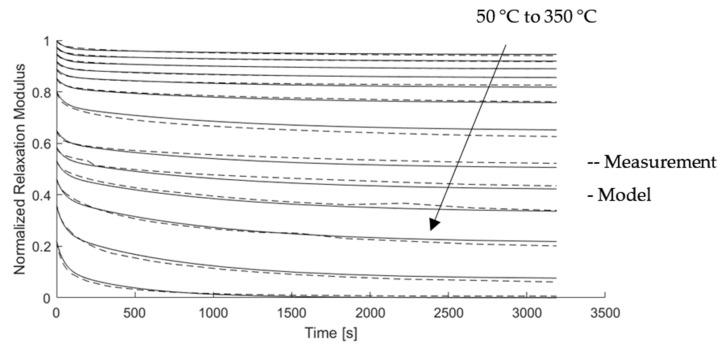
Comparison of the measured relaxation behavior for the developed model in the temperature range of 50 to 350 °C with a relaxation time of 60 min.

**Figure 8 materials-15-06675-f008:**
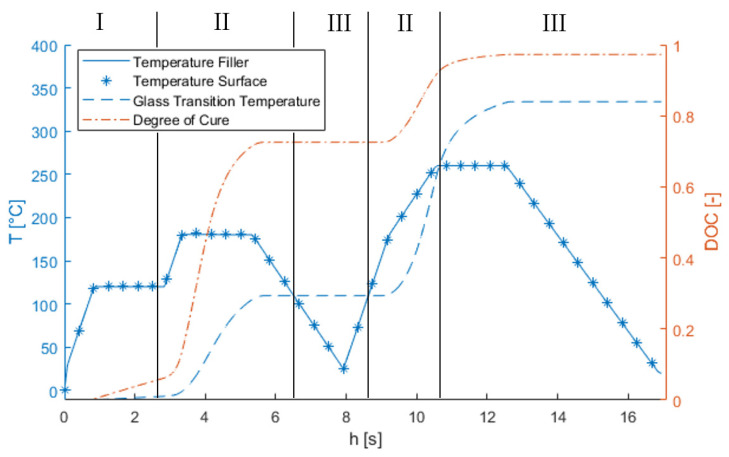
Geometric model and temperature distribution at the end of the process. Visible residual heat in regions with increased part thickness. I–III denote the state of the material in the liquid, rubbery, and solid phase, respectively.

**Figure 9 materials-15-06675-f009:**
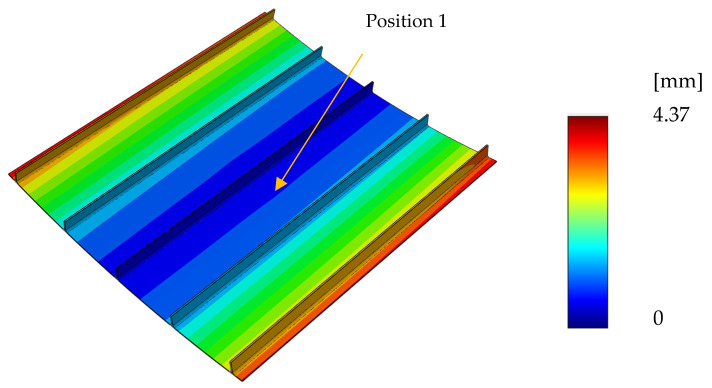
Process-induced distortion at the end of the process.

**Figure 10 materials-15-06675-f010:**
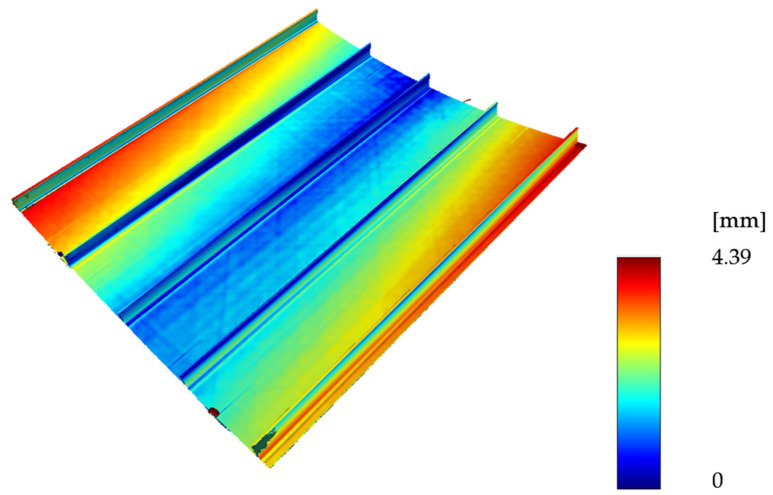
Comparison of the GOM ATOS scan with the reference geometry.

**Figure 11 materials-15-06675-f011:**
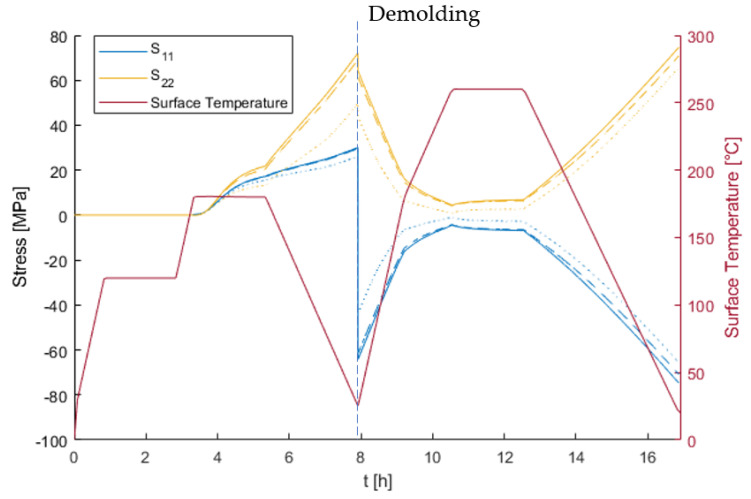
In-plane stress components in the 0° layer. The solid line is the elastic solution, the dashed line is the viscoelastic solution of the fully cured sample, and the dotted line is the viscoelastic solution with cure dependency. Since the layup was chosen to be quasi-isotropic, no remarkable difference was found for all other layers.

**Table 1 materials-15-06675-t001:** Effect of different PES content on the *T_g_* onset of the polymer blend.

Content [phr PES]	*T_g_* Onset [°C]
0	397
5	248.43
10	247.47
15	245.86
20	247.43

**Table 2 materials-15-06675-t002:** Layup.

Element	Layup	Plies	Thickness [mm]
Skin	4 × [45/90/−45/0] + 3 × [0/−45/90/45]	28	1.68
Stringer	[0/−45/90/45]	4	0.24
Blade	5 × [+45/90/−45/0]_S_	40	2.4
Filler	Rolled UD-Laminate		

## Data Availability

The data presented in this study are available on request from thecorresponding author.

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
