# Peer review of "A Manufacturing Process Simulation of Toughened Cyanate-Ester-Based Composite Structures with Respect to Stress Relaxation"

_materials, 2022, doi:10.3390/ma15196675_

Round 1

Reviewer 1 Report

The paper investigates the relaxation behavior of PES modified cyanate esther resin, derived material models and its influence on residual stresses and warpage of a component. Experimental and numerical results are presented. Only marginal corrections should be done before publication:

- Last sentence of abstract: ‘…in a more critical value higher than the material strength’ or similar sounds better.

- Fig. 1: Tg of APC-2-PEEK is wrong. 338°C is the melting temperature!

- Describe the units ‘gsm’ and ‘phr’, please. Especially the second is not common.

- Formula 1: In the text before, ‘E’ is the relaxation modulus while it is later the elastic modulus. Stay consistent in the expressions.

- page 10: ‘The material changes …’ Begin the sentence with a capital letter.

- Fig. 8: Color of Tg and surface temperature can not be distinguished. Furthermore, the surface temperature can not be seen in the diagram. It seems to be congruent with the filler temperature. Try to make it better visible.

- Fig. 9: Check format and value of max. Deformation in the figure caption.

- Fig. 11: Colors of surface temperature and S22 are very similar. Please change it.

- Some figure caption are shifted to the next page.

Reviewer 2 Report

The paper studies an important topic of time-dependent behaviour of modern materials. This is more a technical report than a purely scientific paper. However, the presented methodology and the results should be of interest for readers of the Materials journal. Thus the Reviewer recommends publication of the paper.

There are some minor things that should be fixed anyway: references should be more complete (the generalised Maxwell model should be cleanly referenced and the used FEA software, too). Some more details about hte FEA should be given (types of finite elements used by the software, for example).

It is also recommended to fix the typing errors which are still present in the paper.

Reviewer 3 Report

My main remarks are as follows:

- The authors should situate their study in relation to other research works published in the literature on the effects of stress relaxation in composite materials, in particular in composite materials made of thermosetting matrix toughened with thermoplastic nodules. What approaches have already been published to deal with this problem? What is the real novelty of this study?

- I do not understand why in table 1 there are several values of Tg depending on the content of PES. In my opinion, there are only two Tg: one for the pure matrix (i.e. the pure CE, around 397°C) and other for PES (around 247°C). However, the Tg of PES is better detected for high contents of PES.

- All quantities and parameters used in the equations must be defined in the text. As an example, what is the meaning of Etot? E0? Tref? Tg0? Tg1? Please, define.

- I do not understand why there are no units given for parameters from r1 to r18. However, r1 and r2 are expressed in seconds, and T is expressed in °C or in K…

- There is a mismatch between the numbers of the equations given in the text and in table A1. Please, correct.

- Modeling by equations 2 to 5 involves 18 adjustable parameters. How this set of parameters was obtained? By using an optimization method? Which one? Are we sure that there is not another set of values allowing the same modeling?

- Do these parameters have a physical meaning or are they purely empirical parameters? Could values for these parameters be found in the literature in other research works? If so, are they very different?
